# A Novel Compound Nonsense Variant in *CYP27B1* Causes an Atypical Form of Vitamin D-Dependent Rickets Type 1A: A Case Report of Two Siblings in a Mexican Family

**DOI:** 10.3390/diseases12100248

**Published:** 2024-10-11

**Authors:** Jaime Toral López, Cesar Candia Tenopala, Alix Daniela Reyes Mosqueda, Miguel Ángel Fonseca Sánchez, Luz María González Huerta

**Affiliations:** 1Department of Medical Genetics, Centro Medico ISSEMYM Ecatepec, Ecatepec 55000, México State, Mexico; jaimetor77@gmail.com; 2Department of Pediatric Endocrinology, Centro Medico ISSEMYM Ecatepec, Ecatepec 55000, México State, Mexico; mdcandiacesar@icloud.com; 3Department of Diagnostic and Therapeutic Imaging, Centro Medico ISSEMYM Ecatepec, Ecatepec 55000, México State, Mexico; alixdanielare@gmail.com; 4Department of Investigation, Hospital General de México “Dr. Eduardo Liceaga”, México City 06720, Mexico; miguel.fonseca@salud.gob.mx; 5Laboratory of Molecular Biology, Section of Investigation, Department of Genetic, Hospital General de México “Dr. Eduardo Liceaga”, México City 06720, Mexico

**Keywords:** vitamin D-deficient rickets type 1A, *CYP27B1* gene, compound nonsense heterozygous, hypophosphatemia, atypical, calcitriol

## Abstract

**Background:** Vitamin D-dependent rickets type 1A (VDDR1A) is a rare autosomal recessive disorder caused by pathogenic variants in the *CYP27B1* gene, typically characterized by growth failure, rickets, leg bowing, fracture, seizures, hyperparathyroidism, hypocalcemia, high-alkaline phosphatase, high or normal 25(OH)D3, and low 1,25(OH)2D3. **Methods:** We studied two siblings in a Mexican family with an atypical form of VDDR1A. In addition to the typical features of VDDR1A, the proband showed cafe au lait spots, small teeth, and grayish sclera, with hypophosphatemia, normocalcemia, and normal 25(OH)D3; the proband’s brother showed grayish sclera. The proband underwent next generation sequencing. Sanger sequencing was performed in the proband, his brother, the parents, and 100 healthy controls validate the detected variant. **Results:** Both brothers presented with a recurrent variant NM_000785.3; c.1319_1325dupCCCACCC and a novel nonsense variant NM_000785.3; c.227G>A in the *CYP27B1* gene. **Conclusions:** Calcitriol treatment had a better response in proband´s younger brother. We describe the first Mexican family with an atypical form of VDDR1A associated with a novel nonsense variant, the results contribute to the phenotypic spectrum and increase the pool of pathogenic variants in *CYP27B1*. Data suggest that nonsense-truncating variants play a significant role in the severity of VDDR1A.

## 1. Introduction

Vitamin D-dependent rickets type 1A (VDDR1A; OMIM 264700) is a heterogeneous autosomal recessive disorder caused by pathogenic variants in the cytochrome P450 family 27 subfamily B member 1 gene (*CYP27B1*; OMIM 609506) (NM_000785.3) on chromosome 12q13, which encodes a polypeptide of 508 amino acids—the enzyme 25-hydroxyvitamin D3-1α-hydroxylase (NP_000776.1) [1,2]. The *CYP27B1* gene defect results in the poor conversion of 25-hydroxyvitamin D3 [25(OH)D3] into the active form 1,25-dihydroxyvitamin D3 [1,25(OH)2D3] at the renal level [3]. About 57% of cases described in the literature have a family history of rickets [3]. VDDR1A manifests at around 6 months to 2 years of age, with similar symptoms to nutritional vitamin D-deficient rickets, such as short stature, deformities, bone fractures, hypotonia/weakness, or seizures. Radiographic images show rickets [4,5]. Biochemically, VDDR1A presents elevated parathyroid hormone (PTH) and high-alkaline phosphatase (ALP) with normocalcemia or hypocalcemia [6]. The 25(OH)D3 levels can be normal or elevated, while 1,25(OH)2D3 can be normal or low. Treatment consists of calcitriol, calcium, and/or phosphate supplementation [4]. A total of 81 pathogenic variants in the *CYP27B1* gene have been reported, with 28.6% of them being protein-truncating [7,8]. Hypophosphatemia has been constantly observed in VDDR1A cases [1,9,10,11,12], but it has not been considered relevant for the initial diagnosis, thus confusing inexperienced doctors and delaying diagnosis [1]. Hypophosphatemic rickets has been linked to seven gene mutations [13]. In this study, we describe two brothers with a novel compound nonsense variant in the *CYP27B1* gene associated with the atypical features of VDDR1A. Truncating variants play a significant role in the severity of VDDR1A. This study emphasizes the importance of early diagnosis and treatment to limit complications in patients.

## 2. Case Reports

### 2.1. Patient 1

The 6-year–10-month-old Mexican proband is the first child of healthy, young, unrelated parents at delivery. The family members had no history of VDDR1A or malformations. The mother denied exposure to teratogens or diseases during pregnancy. The proband was born by vaginal delivery at 39 weeks of gestation, with a weight of 3260 g (25th percentile) and a length of 51 cm (75th–50th percentile). Occipitofrontal circumference (OFC) was not registered. The Apgar scores were 8 and 7 at 5 and 10 min. His language and motor development were normal. At 3 months of age, he presented with a fracture in the tibiae, which was treated with conservative management. At 1 year and 3 months of age, he presented with a fracture in the right femur due to a fall from a height of 80 cm and was treated surgically.

X-rays showed a thickened short rib, a bilateral cystic femoral head, shortening of long bones with decreased density, irregular widening of the metaphysis, and curved diaphysis (Figure 1A–D).On the last physical examination at 6 years and 10 months of age, the proband had a weight of 19,250 kg (10th centile), length of 103 cm (<3rd centile; −5.6 standard deviation [SD]), and OFC of 53.5 cm (75th centile). He had frontal bossing, midface hypoplasia, blue-gray sclera (Figure 2B), small teeth, slight pectus carinatum, a widening right wrist, and bilateral genu varum. Café au lait spots greater than 1 cm were observed on the thorax and abdomen. External genitalia were Tanner I. 

An initial biochemical study at 1 year and 1 month of age showed hypophosphatemia, normocalcemia, and normal 25(OH)D3 levels; a diagnosis of hypophosphatemic rickets was suspected. More examinations at 1-year–2-months old showed high levels of PTH and ALP, while the 1,25(OH)2D3 level was low. Thus, a diagnosis of vitamin D-dependent rickets was made. The proband began treatment with calcitriol at a dose of 0.5 μg/day and phosphate supplementation at a standard dose. After two months of treatment (at 1-year–4-months old), phosphorus and 1,25(OH)2D3 parameters returned to normal levels, calcium was low, and PTH and ALP remained high. Calcium supplementation was added at a dose of 500 mg/day. The genetic study at 2 years and 8 months of age confirmed the VDDR1A diagnosis. The biochemical parameters showed normal calcium, phosphorus, and 1,25(OH)2D3, while PTH and ALP remained high. The calcitriol dose was changed to 1.5 μg/day. At 5 years–7-months old, PTH normalized, and ALP was observed at mildly high levels until the last evaluation (6 years 10-months old). His stature was short with curved tibias. Renal tubular acidosis was ruled out in view of the proband’s negative history of polyuria and normal pH in the blood gas. The abdominal ultrasound and echocardiogram were normal. The biochemical parameters and radiographic changes in the proband before and after calcitriol treatment are shown in Table 1 and Figure 1A–D, respectively.

### 2.2. Patient 2

The proband’s brother, a 3-year-old male, is the second child of healthy parents, aged 28 and 34 years at delivery. The mother’s pregnancy was uneventful, without prenatal exposure to teratogens or maternal illness. He was born by vaginal delivery at 37 weeks of gestation, with a weight of 3135 g (25th–10th centile), length of 52 cm (75th centile), and OFC of 34 cm (25th–10th centile). The Apgar scores were 8^1^ and 9^5^, without special neonatal management. On the last physical examination at 3 years old, he had a weight of 16 kg (75th–50th percentile), length of 90 cm (25th–10th percentile; −1.5 SD), and OFC of 51 cm (50th centile). His motor and intellectual development were normal. Sanger sequencing at 2 months old was positive for the same mutation as his brother. Thus, he was treated with calcitriol at a dose of 1.0 μg/day at 3 months old. At 16 months post-treatment (at 1-year–7-months old), his ALP levels remained slightly above the upper limit, while the calcium, phosphorus, 1,25(OH)2D3, and PTH levels were normal (Table 1). Renal tubular acidosis was ruled out. Radiographic images at 1 year of age showed minimal changes (Figure 1E). At 3 years old, his growth was normal, without fractures or body deformities. The renal ultrasound, thyroid profile, and expanded metabolic screening were normal.

## 3. DNA Samples and Next-Generation Sequencing

Written informed consent was obtained from all participants for genetic analysis and publication of the results. This study required hospital ethics committee approval in accordance with the Declaration of Helsinki and local/national guidelines (approval number: DI/23/501/04/32).

Genomic DNA was extracted from the leukocytes of the patients, the parents, and 100 healthy controls using a blood DNA extraction kit, according to the protocol provided by the manufacturer (Promega Inc., Madison, WI, USA).

Next-generation sequencing analysis was carried out in the proband. To capture the exonic regions of interest, the TruSight One Sequencing panel was used, which included 4811 genes related to various genetic bone disorders or hypophosphatemic rickets. Paired-end libraries were generated, and the Illumina HiSeq 2000 Sequencer (Illumina Inc., San Diego, CA, USA) was used for the massive sequencing of these libraries (2 × 150 bp). A bioinformatic analysis of the obtained data was performed using FastQC for the quality control of the reads, BWA for the mapping of the reads, GATK for the identification of the variants, and SnpEff for the verification of the effect of the variants. Subsequently, the variants in the exonic and splice site regions were prioritized using the Clinvar database, HGMD, OMIM phenotypes for clinical correlation, and Mendelian inheritance patterns. The allelic frequency of the variants was determined using 1000 genomes and genomAD. The in silico prediction of the effect of the variant at the protein level was analyzed using Polyphen-2 (Version 2) and SHIFT X2(Version 1.10).

## 4. Variant’s Validation

Validation of the variants was performed using Sanger sequencing from the proband (patient 1), his brother, their parents, and 100 healthy controls using an automated sequencer (ABI3500; PE Biosystems, Foster City, CA, USA). The primers used for amplifying the *CYP27B1* gene were forward 5′-CAGGTATCCAAGTGTCCGCT-3′ and reverse 5′-GATAGTTTCGGGACCCGCAG-3′ for exon 2 and forward 5′-CACTCTGTGTCACTATGCCAC-3′ and reverse 5-GAAGATTCATTCTACCAGGTC-3′ for exon 8.

## 5. Statistical Analysis

Continuous variables are presented as means ± standard deviations. After verifying a normal distribution, Student’s *t*-test (two-sided) and repeated measures analysis of variance (ANOVA) were used to compare the paired variables of changes in the biochemical parameters before and after calcitriol treatment. A *p* value < 0.05 was considered statistically significant. SPSS version 24.0 statistical software (SPSS Inc., Chicago, IL, USA, 2016) was used for statistical analysis.

## 6. Genetic and Biochemical Results

A previously reported heterozygous pathogenic variant c.1319_1325dupCCCACCC (p.phe443Profs*24) (Figure 2C) was detected in exon 8 [9,11,12,14,15], and a novel heterozygous nonsense variant c.227G>A (p.Trp76*) was located in exon 2 of both brothers (Figure 2D). The father was a heterozygous carrier for the variant in exon 8 (Figure 2E), while the mother was a heterozygous carrier for exon 2 (Figure 2F). The c.227G>A (p.Trp76*) variant has not been described in the literature or in databases or population registries. SnpEff showed a high impact on protein switching, and an allele frequency of 0.0 was obtained by 1000 genomes and genomAD. In silico predictors suggested that c.227G>A (p.Trp76*) was a deleterious change. This change was not found in 100 healthy, normal controls. The truncation mutation p.Trp76* was situated in the N-terminal end of the CYP27B1 protein, losing the entire region of the binding pocket to the substrate downstream [16].

The biochemical parameters of the proband and his brother were compared before and after calcitriol treatment at 53 and 16 months of follow-up, respectively. Significant statistical differences were observed for calcium, phosphorus, ALP, and PTH, while no significant differences were found for 25(OH)D3 and 1,25(OH)2D3 in the proband when analyzed using repeated measures ANOVA. However, when Student’s *t*-test for paired samples was applied, statistically significant differences were found for 1,25(OH)2D3. In the proband’s brother, statistically significant differences were found for calcium, PTH, and 1,25(OH)2D3 using Student’s *t*-test for paired samples (Table 1).

## 7. Discussion

This study describes two Mexican brothers with VDDR1A and a novel c.227G>A (p.Trp76*) heterozygous nonsense variant along with a recurrent c.1319_1325dupCCCACCC (p.phe443Profs*24) in the *CYP27B1* gene. Our proband presented atypical café au lait spots, small teeth, bossing frontal, and grayish sclera. Initially, hypophosphatemia, normocalcemia, and normal 25(OH)D3 caught our attention. The proband showed a partial response, while his youngest brother had a good response to calcitriol treatment. A brief literature review of VDDR1A cases with *CYP27B1* pathogenic variants clinically showed 59% failure to thrive/growth retardation, 38–41% delayed motor milestones/inability to walk, 13% poor feeding, 27% leg bowing, 16.5% seizure, and 13.6% fracture. They also biochemically showed 75% hypocalcemia, 92% high ALP, 42% high 25(OH)D3, 45% low 1,25(OH)2D3, 91% high PTH, and 68–83% hypophosphatemia [1,9,10,11,12]. The café au lait spots seen in our proband were previously reported in an 11-month-old girl. The authors argued that the 25(OH)D3 level was too low [17], but it was considered normal in our proband. The small teeth observed in our proband were also documented in 81% of adults and 40% of children in relation to delays in dentition [11] or enamel hypoplasia [10]. Bossing frontal was observed in one case [7]. The blue-gray sclera seen in our proband could be considered relevant because it has not been reported in VDDR1A cases and can be confused with imperfect osteogenesis. It is noteworthy that hypophosphatemia, an initial biochemical result in our patients, has been observed constantly in retrospective studies of VDDR1A cases [11] but has not been significantly considered an initial indicator of VDDR1A, thus often delaying diagnosis [1]. Hypophosphatemia in VDDR1A results from elevated PTH and renal phosphate excretion [9]. Other pathogenic variants in *PHEX*, *FGF23*, *CLCN5*, *DMP1 ENPP1 SCL34A3*, *VDR*, and *CYP2R1* genes can manifest hypophosphatemic rickets [13]. VDDR1A has complete penetrance, but clinical heterogeneity can be observed with the same mutation in *CYP27B1* [8,11,18]. Interestingly, some cases recovered from the loss of CYP27B1 enzyme function, probably due to the 1α-hydroxylase activity exerted by a non-CYP27B1 enzyme [15,16].

To the best of our knowledge, around 81 pathogenic variants (58% missense, 23% frameshift, 9.8% nonsense) have been identified in the *CYP27B1* gene in patients with VDDR1A worldwide, with 64.4% homozygous and 38% compound heterozygous results [7,8]. This is the first time the recurrent pathogenic variant c.1319_1325dup has been reported in the Mexican population. Previously, the case of a Mexican child with typical features was described but did not include genetic studies [9]. Table 2 presents previous cases in a homozygous or compound state with other variants [3,7,10,12,15,16,19]. To the best of our knowledge, this study is the first to report the combination of the variant c.1319_1325dup compound with a nonsense variant, both of which result in a truncating protein. This is explained by RNA degraded by the nonsense mechanism, causing severe inactivation of the 1α-hydroxylase enzyme [5]. Our proband with the heterozygous c.1319_1325dup (p.phe443Profs*24) and the c.227G>A (p.Trp76*) variants presented with hypophosphatemia, normocalcemia without seizures, fractures, and severe growth retardation (height −5.6 SD at 6 years–10-months old). Café au lait spots and small teeth, which are atypical features, were also seen, and blue-gray sclera was observed for the first time, supporting the clinical heterogeneity for these variants.

Doses of calcitriol at 0.5–2 μg/day or 0.008–0.40 μg/kg/day have been used to treat VDDR1A [11,16,20]. Our proband was treated with calcitriol at doses of 0.5–1.5 μg/day. After 53 months of follow-up (at 5 years–7-months old), all biochemical parameters normalized, except for ALP, which remained slightly high (Table 1). A similar behavior was seen in the proband’s brother after a 16-month follow-up (at 1-year–7-months old), which is indicative of pathological activity. However, in our proband, radiographic images showed partial improvement (Figure 1A–D), and he clinically remained with short stature and a curved tibia. In the proband’s brother, imaging studies showed mild changes in cupping and fraying without widening (Figure 1E), and stature was normal (−1.5 SD), without deformities or nephrocalcinosis. Our proband had good compliance with his treatment. The partial response to treatment could be explained by the delay in management and the presence of truncated variants, as previously observed. Adults or children older than 12 years with the truncating variant and delayed treatment were more severe than those without this variant type, which led to permanent short stature and deformities [1,11]. In another study, 12 children who presented truncating by the frameshift variant in at least one allele were observed with delayed growth when they were treated late and did not have correct compliance [19]. In the treatment of a 13-month-old girl, a good biochemical response at 10 months post-treatment was observed, but she presented with multiple fractures; no growth or deformity data were reported [7]. In our second case, the diagnosis at 3 months and early treatment helped obtain a good biochemical response and limit rickets without delayed growth, fractures, or deformities. Therefore, according to this study and previous research, the truncated variants suggest a more severe phenotype, but an early diagnosis, early treatment with correct compliance, and an adequate dose are important factors that must be considered for a better health prognosis in patients with VDDR1A.

## 8. Conclusions

In this study, we describe a Mexican VDDR1A family with atypical features associated with a novel compound nonsense variant. The results contribute to the phenotypic spectrum and increase the pool of pathogenic variants in *CYP27B1*. Truncating variants play a significant role in the severity of VDDR1A. In patients with rickets, initial hypophosphatemia, normocalcemia, and normal 25(OH)D3, *CYP27B1* should be investigated as a possible diagnosis. These results can be useful in the genetic counseling of patients. We emphasize the importance of early diagnosis and treatment to limit complications in patients.

## Figures and Tables

**Figure 1 diseases-12-00248-f001:**
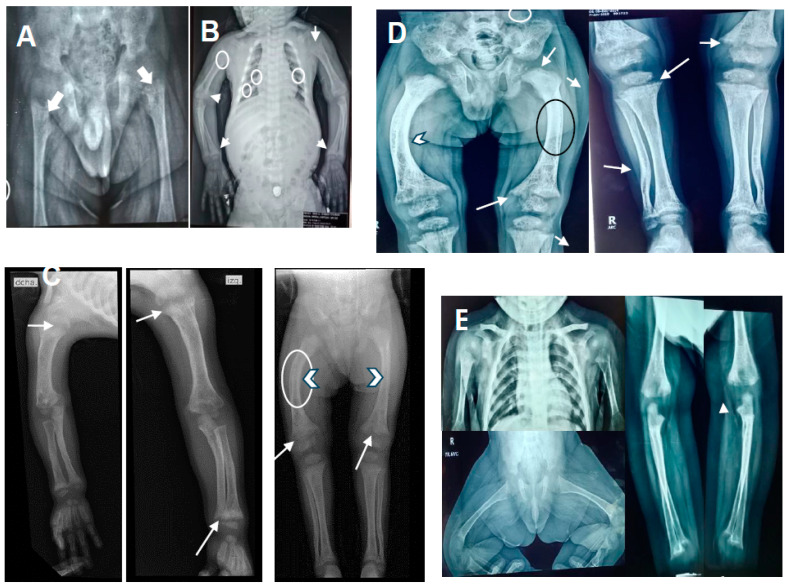
X-rays of the proband and his brother before and after calcitriol treatment. (**A**) The proband at 10 months old with discrete cupping and widened metaphysis (white arrows). (**B**) The proband at 2 years–6 months old (18 months post-treatment) with a wide anterior rib ridge, irregularity of coastal bodies associated with osteopenia (small white circles), humerus with diffuse osteopenia (large white circle), and increased amplitude of the growth plate at the radius and ulna levels with frayed and cupped metaphysis and membranous ossification (white arrows), consistent with rickets. (**C**) The proband at 3 years old and (**D**) at 4 years–6 months old shows partial recoveries with a diaphyseal slope (white arrowhead) and cupping, widening, and fraying in the metaphysis of long bones (white arrows). Qualitative improvement in bone density was observed (black circle). (**E**) X-ray of the proband’s brother at 9 months post-treatment (1 year old). Minimal changes in the bone density and cupping or fraying without widening the metaphysis of the humerus, femur, radius, and ulna (white arrows) are observed.

**Figure 2 diseases-12-00248-f002:**
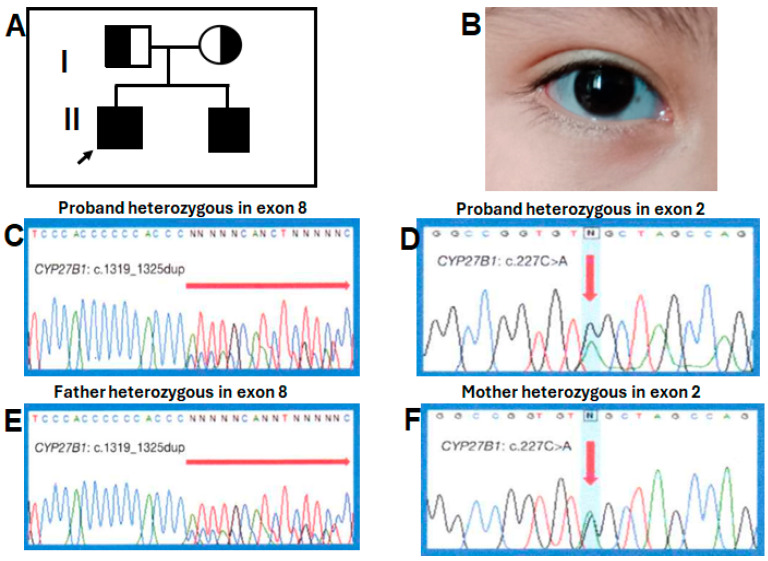
(**A**) Pedigree of the family illustrating the carrier status of the parents, the proband (arrow), and his affected younger brother. (**B**) The blue-gray sclera in our proband. (**C**,**D**) Partial electropherogram of the recurrent heterozygous variant in exon 8 and the novel heterozygous pathogenic variant in exon 2 in the proband, respectively. (**E**) Partial electropherogram of the father with the c1319_1325dupCCCACCC (p.phe443Profs*24) variant in exon 8 and (**F**) the novel c.227C>A (p.Trp76*) variant in exon 2 of the *CYP27B1* gene in the mother.

**Table 1 diseases-12-00248-t001:** Changes in biochemical parameters before and after calcitriol treatment in our VDDR1A patients.

Proband (Patient 1)							Proband’s Brother (Patient 2)		
Features	Reference Ranges	Level Pre-Treatment	After 2 at 18 Months of Treatment	After 18 at 53 Months of Treatment	After 53 Months of Treatment	F of ANOVA Test, *p* Value	Pre-Treatment	16 mo Post-Treatment	Student’s *t*-Test, *p* Value
Age, y (year), mo (months)		1y-1 mo	1y-4 mo at 2y-8 mo	2y-8 mo at 5y-7 mo	5y-7 mo to 6y-10 mo		<3 mo	3 to 1y-7 mo	
Calcium, mg/dL, mean ± SD	8.3–10.6	8.3	8.14 ± 0.6	8.6.0 ± 0.2	9.5 ± 0.4	<0.001 *	8.0	9.0	0.02 **
Phosphorus, mg/dL, mean ± SD	2.4–5.1	2.1	3.1 ± 1.1	3.2 ± 0.5	4.8 ± 0.7	<0.006 *	3.8	4.2	0.59
AP, UI/L, mean ± SD	45–129	9200	5098 ± 2108	2635 ± 300	171 ± 62	0.02 *	1336	682	0.05
PTH, pg/mL, mean ± SD	10–88	1350	702 ± 185	289 ± 137	60 ± 27	<0.001 *	314	67	<0.001 **
25(OH)D3, ng/mL, mean ± SD	30–100	32.9	73.9 ± 13	51.9 ± 12	62 ± 15	0.06	81	91	0.62
1,25(OH)2D3 pg/mL, mean ± SD	19.6–54.3	6	36.7 ± 14	34.8 ± 12	45.3 ± 4	0.12 (<0.001 **)	9	43	0.01 **
Height, cm		67	67 at 77	77 at 92	92 at 103	NA	59	79	
Height, SD		−3.2	−4.3 to −5.3	−5.3 to −6.6	−6.6 to −5.6	NA	−0.6	−1.3	
Calcitriol dose, μg/day		NA	0.5	1.5	1.5	NA	NA	1.0	

Average of biochemical parameters was analyzed with ANOVA of repeated measures; (*) *p* statistically significant; SD: standard deviation; NA, not applicable; (**) *p* statistically significant in the pre- and post-treatment with paired Student’s *t*-test.

**Table 2 diseases-12-00248-t002:** Genetic and clinical response to calcitriol treatment in VDDR1A cases involving the c.1319_1325dupCCCACCC homozygous or compound with other variants in the *CYP27B1* gene.

Cases/Captation Age	DNA Mutation	Exon	Amino Acid Change	Phenotype	Clinical Response at Calcitriol Treatment	Author
2 cases (4 mo). 2 cases (18 and 19 mo). 1 case (INR)	c.1319_1325dupCCCACCC. c.1319_1325dupCCCACC. c.1166G>A; c.1079 C>A	8 8 7 6	p.Phe443Profs*24. p.Phe443Profs*24. p.Arg389His; p.Ser360*	Severe hypocalcemia and seizure. Delay in walking and mild hypocalcemia. Bowed legs.	INR	[16]
1 case (13 mo)	c.1510C > T.	9	p.Q504*.	Multiple fractures, bossing frontal, and classic VDDR1A.	Good biochemical response at 10 mo post-treatment. Growth and deformity NR	[7]
Case 8 (INR) Case 7 (INR)	c.574A>G; c.1319_1325dupCCCACCC. c.1319_1325dupCCCACCC	3 8 8	p.K192E; p.Phe443Profs*24. p.Phe443Profs*24.	Mild phenotype. Severe phenotype.	INR	[12]
Case 5 (14 mo) Case 8 (24 mo)	c.1319_1325dupCCCACCC. c.1319_1325dupCCCACCC	8 8	p.Phe443Profs*24. p.Phe443Profs*24.	Growth retardation and hypocalcemia. Inability to walk and mild hypocalcemia.	INR	[10]
2 cases (INR)	c.1319_1325dupCCCACCC.	8	p.Phe443Profs*24.	Hypocalcemic seizure, severe growth retardation, walking difficulty, and skeletal deformities.	INR	[15]
3 cases (4–18 mo)	c.1319_1325dupCCCACCC.	8	p.Phe443Profs*24.	Hypocalcemic seizures in infancy, rickets, dental anomalies, and fractures.	Good biochemical response, two patients > 12 years persisted deformity	[3]
12 cases	c.1319_1325dupCCCACCC (Eight cases involving this variant)	8	p.Phe443Profs*24.	Delayed walking and severe growth retardation.	Good rickets and biochemical response (6 mo to 15.6y of follow-up), and 58% of patients remained with short stature	[19]
Case 1 (proband; 13 mo) Case 2 (Brother; 2 mo)	c.227G>A; c.1319_1325dupCCCACCC. c.227G>A; c.1319_1325dupCCCACCC.	2 8 2 8	p.Trp76*; p.Phe443Profs*24. p.Trp76*; p.Phe443Profs*24.	Low-normal calcemia, no seizures, fractures, severe growth retardation, sclera gray, café-au-lait spots, frontal bossing, mild medial facial hypoplasia, and pectum carinatum. Gray sclera.	Good biochemical response, partial to rickets, and bad growth deformity prevention Good biochemical, rickets, growth response, and prevention of deformities	This study

OD: onset of disease, CH: compound heterozygous mutation, INR: information not reported. DAH: attention deficit-hyperactivity disorder, y: year old, mo: months.

## Data Availability

All data generated or analyzed during this study are included in the final published article.

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
