# Peer review of "A Novel Compound Nonsense Variant in CYP27B1 Causes an Atypical Form of Vitamin D-Dependent Rickets Type 1A: A Case Report of Two Siblings in a Mexican Family"

_diseases, 2024, doi:10.3390/diseases12100248_

Round 1
Reviewer 1 Report
Comments and Suggestions for Authors
This thorough and detailed case report of two siblings with Vitamin D-dependent rickets adds to existing knowledge of the clinical findings associated with this rare pathogenic gene variant, which results in defects in the renal enzyme that hydroxylates vitamin D at the 1 alpha position. The paper is useful to clinicians since it presents some previously unreported and novel signs that may help lead to earlier diagnosis, increasing the probability of successful treatment with calcitriol. The paper also will be of great interest to geneticists, endocrinologists, and other scientists through its contribution to the understanding of the pathophysiology of rickets and the importance of enzyme activation of this prohormone.
Comments on the Quality of English Language
The paper is clearly written, well-organized, and easily understood, however, there are some minor deviations from standard English that could be addressed in copy-editing.
Author Response
Reviewer 1
- This thorough and detailed case report of two siblings with Vitamin D-dependent rickets adds to existing knowledge of the clinical findings associated with this rare pathogenic gene variant, which results in defects in the renal enzyme that hydroxylates vitamin D at the 1 alpha position. The paper is useful to clinicians since it presents some previously unreported and novel signs that may help lead to earlier diagnosis, increasing the probability of successful treatment with calcitriol. The paper also will be of great interest to geneticists, endocrinologists, and other scientists through its contribution to the understanding of the pathophysiology of rickets and the importance of enzyme activation of this prohormone.
Answer: Thank you so much
- The paper is clearly written, well-organized, and easily understood, however, there are some minor deviations from standard English that could be addressed in copy-editing.
Answer: Regarding the sending of the English edition, we comment that it has already been sent to edition with a Native English-speaking editor at "Scribendi Customer Service" with Order #: 1018791, Access code: Q4n56FH6yCkc, whose corrections have already been incorporated into the text.
Reviewer 2 Report
Comments and Suggestions for Authors
General comment: The authors describe a pathological variant (mutation) of the CYP27B1 gene in two siblings in a mexican family. This new variant results in Vitamin D -Dependent Rickets when present together with a known mutation (compound mutation). The authors describe clinical symptoms and therapy with calcitriol . The papers adds new information on vitamin D-dependant rickets type 1 and discusses differential diagnosis.
Specific comment: The paper ist well written and provides all information.
Comments on the Quality of English Language
Only minor corrections necessary.
Author Response
- General comment: The authors describe a pathological variant (mutation) of the CYP27B1 gene in two siblings in a mexican family. This new variant results in Vitamin D -Dependent Rickets when present together with a known mutation (compound mutation). The authors describe clinical symptoms and therapy with calcitriol. The papers adds new information on vitamin D-dependant rickets type 1 and discusses differential diagnosis.
Specific comment: The paper ist well written and provides all information.
Answer: Thank you so much
- Comments on the Quality of English Language. Only minor corrections necessary.
Answer: Regarding the sending of the English edition, we comment that it has already been sent to edition with a Native English-speaking editor at "Scribendi Customer Service" with Order #: 1018791, Access code: Q4n56FH6yCkc, whose corrections have already been incorporated into the text.
Reviewer 3 Report
Comments and Suggestions for Authors
This is an intriguing study which throws considerable light on the genetic defect range which causes low or none activity of the CYP27B1 vitamin D 1-hydroxylase enzyme. It is very helpful to have a summary of other publications which report other genetic variances causing the same defect and resulting in vitamin D-dependent rickets. There is one small difficulty which would improve understanding of the study.
At line 57 the fist case is described as a 6-year-10 month-old individual. Then at lines74, 76, 79, 82, 84 and 85, the individual is described at ages of 13, 14, 16, 32, 67 and 82 months of age. This is confusing. Why is the subject introduced at 6-years, 10 months of age when the following information seems to relate to earlier ages? Were these observations at different months following on from 6-years and 10 months of age?
Again the second subject at line 90 is described as being 3 years old. Then follows observations at different months. Are these observations at intervals of months following the initial point of 3 years of age or are they from age after birth? The description for the timing of both subjects needs to be rewritten to void the current confusion.
Comments on the Quality of English Language
There are several sentences where the structure is incorrect. It would be wise for the manuscript to be revised so that every sentence is written with the correct use of English.
Author Response
This is an intriguing study which throws considerable light on the genetic defect range which causes low or none activity of the CYP27B1 vitamin D 1-hydroxylase enzyme. It is very helpful to have a summary of other publications which report other genetic variances causing the same defect and resulting in vitamin D-dependent rickets. There is one small difficulty which would improve understanding of the study.
- At line 57 the fist case is described as a 6-year-10 month-old individual. Then at lines74, 76, 79, 82, 84 and 85, the individual is described at ages of 13, 14, 16, 32, 67 and 82 months of age. This is confusing. Why is the subject introduced at 6-years, 10 months of age when the following information seems to relate to earlier ages? Were these observations at different months following on from 6-years and 10 months of age?
Answer: To avoid confusion, we have now changed the age of the patients described in months of age to years and months of age.
- Again the second subject at line 90 is described as being 3 years old. Then follows observations at different months. Are these observations at intervals of months following the initial point of 3 years of age or are they from age after birth? The description for the timing of both subjects needs to be rewritten to void the current confusion.
Answer: To avoid confusion, we have now changed the age of the patients described in months of age to years and months of age.
- Comments on the Quality of English Language. There are several sentences where the structure is incorrect. It would be wise for the manuscript to be revised so that every sentence is written with the correct use of English.
Answer: Regarding the sending of the English edition, we comment that it has already been sent to edition with a Native English-speaking editor at "Scribendi Customer Service" with Order #: 1018791, Access code: Q4n56FH6yCkc, whose corrections have already been incorporated into the text.